# Metagenomic analysis of sewage for surveillance of bacterial pathogens: A release experiment to determine sensitivity

**Simon Kohle, Thomas N. Petersen, Håkan Vigre, Markus Hans Kristofer Johansson, Frank M. Aarestrup***

Research Group for Genomic Epidemiology, DTU-Food, Technical University of Denmark, Kongens Lyngby, Denmark

* fmaa@food.dtu.dk

**Data Availability Statement:** The raw data are available at https://www.ebi.ac.uk/ena/browser/view/PRJEB41153.

## Abstract

Accurate monitoring of gastro-enteric and other diseases in large populations poses a challenge for public health management. Sewage represents a larger population, is freely obtainable and non-subject to ethical approval. Metagenomic sequencing offers simultaneous, multiple-target analysis. However, no study has demonstrated the sensitivity of metagenomics for detecting bacteria in sewage. In this study, we spot-released $10^{13}$ colony-forming units (CFU) of *Staphyloccus hyicus* (non-pathogenetic strain 842J-88). The strain was flushed down a toilet into the sewer in the catchment area of a public wastewater treatment plant (WWTP), serving a population of 36,000 people. Raw sewage was continuously sampled at the WWTP's inlet over 30- and 60-minute intervals for a total period of seven hours. The experiment was conducted twice with one week in-between release days and under comparable weather conditions. For the metagenomics analyses, the pure single isolate of *S. hyicus* was sequenced, assembled and added to a large database of bacterial reference sequences. All sewage samples were analyzed by shotgun metagenome sequencing and mapped against the reference database. *S. hyicus* was identified in duplicate samples at both of two release days and these sequence fragment counts served as a proxy to estimate the minimum number of sick people or sensitivity required in order to observe at least one sick person at 95% probability. We found the sensitivity to be in the range 41–140 and 16–36 sick people at release days 1 and 2, respectively. The WWTP normally serves 36,000 people giving a normalized sensitivity in the range of one in 257 to 2,250 persons.

## Introduction

Infectious diseases are among the biggest global threats to human health. Accurate surveillance data are essential for monitoring spatial and temporal occurrences of pathogens, but obtaining such data on disease burdens in a population poses a considerable challenge. This may be especially true for self-limiting or intense, short-term diseases, such as campylobacteriosis (2–10

**Funding:** This study was supported by the EU's Horizon H2020 grant VEO (874735) and Novo Nordisk Foundation (NNF16OC0021856). The funders had no role in study design, data collection and analysis, decision to publish, or preparation of the manuscript.

**Competing interests:** The authors have declared that no competing interests exist.

days) or norovirus infections (2 days), that remain largely unidentified and unreported, due to the ill not seeking medical advice [1, 2].

In recent years, advances have been made in various fields of metagenomic sequencing, including for surveillance of viruses and bacteria [3–7], rapid clinical diagnostics [8, 9] or the monitoring of the resistome [10–12].

Monitoring transmittable diseases through metagenomic sequencing of sewage may offer public health policy makers an advantageous solution [13]. Not only does a sewage sample automatically represent a larger population, it is also easily obtainable and non-subject to ethical approval [6, 13]. In addition, metagenomic sequencing offers simultaneous multi-target analysis [3, 4, 8, 9].

Analyses of sewage has for many years been an integrated part of the surveillance for poliovirus [14, 15] and has shown great promises for surveillance of other pathogens [16–18]. Lately sewage-based surveillance of SARS-CoV-2 have gained major attention [18–20]. Limited information is however, available on the sensitivity of the approach for estimating number of infected people. When releasing $5 \times 10^{10}$ cell culture infectious dose 50% into a sewage system serving 700,000 people, Hovi et al. [21] were able to detect the poliovirus for 4 days in only 400 mL of samples. Based on this it was estimated that monitoring of poliovirus using sewage would have a sensitivity of detecting one infected person in 10,000. However, to the best of our knowledge no study has previously attempted to determine the sensitivity of using metagenomics for detection of bacterial pathogens in terms of number of ill persons.

In this study, we released known quantities of a bacterium not normally found in sewage, namely *Staphylococcus hyicus*, into a toilet, and analyzed sewage from the inlet to the wastewater treatment plant by metagenomics to determine the sensitivity of this approach for surveillance.

## Material and methods

### Overall concept

The basic concept was, to flush known amounts of a bacterial target, normally not present in sewage, down a toilet and then detect and quantify them at the inlet of the corresponding wastewater treatment plant (WWTP), serving a known number of people. Continuous sampling in half- and one-hour intervals were conducted at the WWTP from one hour before until six hours after release to determine the fragment counts of the target organism from first appearance through the peak concentration of target bacteria until the end of the tail where all has been flushed through the sewage system.

The samples were quantitatively analyzed by metagenomics sequencing and the release experiment repeated once under equal conditions. From the released amount of CFU, we estimated the minimum number of ill persons required in order to observe at least one via metagenomic mapping/alignment of sequence reads to a reference sequence database including the target organism.

### Release organism

*Staphylococcus hyicus* is a Gram-positive, non-motile coccus with an optimal growth temperature of 37° C. The selected strain, 842J-88, was shown to be avirulent for the skin diseases some *S. hyicus* strains cause in various farmed animals [22]. The incubation temperature was set to 37° C.

Duplicate growth assays were conducted to indicate the effective enumeration time from high-ratio inoculation to maximum titre. Cell counting and photo spectrometry were applied, to establish correlations between the cell titre and absorbance at 600 nm (S1 Fig in S2 File). This allowed swift monitoring of growth and determination of the final cell titre prior to

release. To do so, overnight cultures were added to growth medium at room temperature (1:100) and the cell concentration measured repeatedly over six hours. The *S. hyicus* organism was grown aerobically and with agitation at 37° C and quantified by culturing 10-fold dilution series on Columbian blood agar plates (Statens Serum Institute, Denmark) at 37° C.

## Purification prior to enrichment for release and sequencing

The bacterial culture was purified prior to reference sequencing and enrichment for final release. Ten-fold dilution series were plated on the respective growth media (S1 Table in S1 File) and incubated for 24–72 hours at 37° C. The plates were examined for colony morphology and color. The cells of a single colony were microscopically examined for shape, motility and homogeneity, before re-plating them in a 10-fold dilution series. The process was repeated three times. No contamination was ever detected. Aliquots from an overnight culture of the last colony picking were collected for reference sequencing.

**Enrichment scale-up.** At the release day, 182 litres (L) of high-titre stock were produced from a continuous, pure culture in four scale-up steps over 26 hours. First, 600 mL of sterile medium at room temperature in a Schott flask were inoculated (1:10) and incubated for six hours at 37° C and 180 rounds per minute (rpm; IKA KS 4000 i control, IKA-Werke GmbH & CO. KG, Germany). The lid was replaced by a 0.2-μm filter paper, held in place by a rubber band, to enable gas exchange. Second and third, three and 24 Schott flasks with 1.8 L of medium at room temperature were inoculated (1:10) with the previous enrichment and incubated in the same fashion. Fourth, the enriched contents of the 24 Schott flasks were pooled and homogenized in two 27-L buckets. Sterile, prewarmed medium from another 75 1.8-L Schott flasks was evenly divided into 22-L buckets. They were then inoculated (1:3.8) with the pooled, third enrichment and incubated for 6–8 hours at 37° C and 300 rpm with lids ajar (S4 Fig in S2 File). For the final enrichment step, a walk-in incubator with ample shelf-space was utilized (S2-S4 Figs in S2 File). Magnetic stirrers were placed amidst wooden jigs for the 22-L buckets to stand on. Crucially, to achieve and maintain the target incubation temperature of 37° C, the medium was preheated for 24 hours as well as transferred and inoculated inside the incubator with a minimum of door openings.

On both release days, the bacterial cultures were removed from the incubator at 8 AM, loaded-up for transport to the release site by 9 AM and ready to be released at 11 AM. Prior to loading, each bucket was sampled for determination of cell titre by photo spectrometry. Triple-readings were taken at 600 nm and the overall mean extrapolated by the trend-line formula in S1 Fig in S2 File to attain the total cell count.

The WHO defines diarrhea as "the passage of three or more loose or liquid stools per day" [23]. The definition is based on the number of stools and their consistency, rather than on their volume [24]. Data on amount of feces excreted daily are surprisingly scare. The average healthy person excretes less than 200 g per day, but with large variations [24, 25] and is normally increased in people with diarrhea. For the present study, an amount of 200 g of feces was assumed for the main motion of an acute diarrheic. A typical fraction of bacterial biomass in healthy feces is 5–13 percent [24]. This presumably declines with a diarrheic colon's inability to reabsorb liquids. The CFU of diarrheal pathogens also varies a lot [26–28]. For this work, a concentration of $10^6$ CFU per gram was assumed for diarrhea-inflicting bacterial agents, such as *Campylobacter* spp., *Salmonella* ssp. and *Vibrio cholera*. We can therefore approximate a total number of CFU to number of "sick persons".

## Release

The bacterial target was released into the sewage system by flushing it down a toilet in Kokkedal, Denmark. The basic procedure was to pour all buckets into the bowl, followed by the cold

**Table 1. CFU released is the number of colony forming units and numbers in the last column are calculated.**

| Date | synonym | Time | CFU [$10^{13}$] | Estimated number of sick persons |
|---|---|---|---|---|
| 26 Sep 2019 | Day 1 | 11:07–11:22 | 1.66 | 83,146 |
| 3 Oct 2019 | Day 2 | 11:35–11:50 | 1.91 | 95,525 |

rinse of lids and buckets, then flush several times. This flushing time was 15 minutes. Release times are shown in Table 1.

The weather on both release days was overcast, 11–15˚ C and 2–10˚ C on release day 1 and 2, respectively, with occasional light showers during the last sampling period on release day 2.

## Hørsholm waste water treatment plant

Hørsholm WWTP, is a conventional sewage treatment facility. After initial screening for foreign objects and grit removal, the raw sewage is processed by two rounds of sedimentation and sludge treatment. The plant was designed for a maximum capacity of 50,000 people. At the time of these experiments, it served the equivalent of 36,000 people from private households and minor industry.

On weekdays,–such as the Thursdays of the experiments–the WWTP typically encountered peak flow rates early in the morning and in the evening, as commuters departed for and returned from work. We observed a max flow rate around 10 and 10:30 as shown in S1 and S2 Tables in S1 File.

The more densely populated residential area, from which the bacterial targets were released, accounted for 10,000 of the plant's effective processing volume of 36,000 people.

There were two pump stations between the release point and the entry of the WWTP to provide lift and maintain gravitational flow. The pump stations were circular basins (3 m in diameter, 1.5 m deep), let into the ground. The first pump ran about every 15 minutes during the day and every 30 minutes at night. The second pump ran every minute. The arrival time of the bacterial target at the sampling point, post release, was 30–60 minutes, based on WWTP's engineers' best estimate, based on distance, gravity flow rate and usual activity of the two pump stations at time of the experiments.

## Sampling

Samples of raw sewage were collected at the inlet, after the rake and the sedimentation basin for grit removal. A custom-built, continuous sampling station was established next to the plant's inlet flow meter and own, non-continuous sampling station (S5 Fig in S2 File).

The sampling station was manufactured from a 1,200-L plastic gardening box (146x125x82 cm, Keter Max, Knud Larsen Byggecenter, Denmark), painted white to reflect sunlight. It held a 135-L refrigerator, a peristaltic pump as well as sampling and hygiene materials (S6 Fig in S2 File). The fridge and reflective painting served to hold the 27-L primary sampling buckets and numerous sub-sampling vessels at <5˚ C, even on a sunny day exceeding 20˚ C. The temperature inside the fridge was continuously monitored with a thermal probe. The setup also withstood torrential horizontal rainstorms without break-ins of water.

A peristaltic pump (Ismatec BVP-Standard with Masterflex L/S Easy-Load II pump head, Cole-Parmer GmbH, Germany) with silicone tubing (4.8 mm inner diameter, 1.6 mm wall thickness) was used for sampling. Total lift height and tubing length were 1.5 and 4 m, respectively. A sampling lance, consisting of a galvanized, 1/2-inch metal pipe with detachable sampling head, was used to sample mid-inlet stream, about 25 cm below the surface (S7 and S8 Figs in S2 File).

The sampling head was constructed from a fine mesh, stainless steel kitchen appliance sieve (20 cm across) to filter out coarse debris (S9 Fig in S2 File). It successfully prevented clogging of the silicon tubing–even over a pilot test period of three days–and aided delivery of comparatively low-particle sewage samples (S9 Fig in S2 File).

Sampling took place from one hour prior to six hours post release; a one-hour interval prior release, followed by four half-hour and four one-hour intervals. The overall pump and tubing setup was calibrated on each release day to a sampling rate of 20 L per hour. The consistency of the sampling volume was monitored by measuring the height of the sewage level in the bucket. New, washed and autoclaved tubing was applied each day. Freshly washed and disinfected buckets were used for every sampling interval.

At the end of each sampling interval, the pump was briefly stopped. The primary sampling bucket was exchanged and the pump restarted within a minute. The sample was homogenized for 30–60 seconds with an electric drill and a paint stirring attachment. Four subsamples of 0.25 L were drawn on site and stored in the fridge at 5° C.

The subsamples were regularly transported back to the laboratory and processed within 4 hours post sub-sampling at the WWTP. Two of the subsamples were immediately frozen at -20° C. The third was combined with 20 percent of glycerol and also frozen at -20° C. The fourth subsample was added to 1.5 L of growth medium at room temperature and incubated aerobically at 37° C and 180 rpm. After 12 hours, a subsample of 0.25 L was drawn and frozen at -20° C. All frozen samples were transferred from -20° C to -80° C after 24 hours.

Data on the throughput of sewage during the time of the experiments was obtained through the WWTP's flow meter, situated right next to the sampling station.

## DNA-sequencing and metagenomic analyses

The DNA was extracted as described by Knudsen et al. [29], with the exception that the entire pellet from 0.25 L of sewage was extracted. Both the weight of the pellet as well as the DNA concentration of the eluate were recorded. One nanogram (ng) of DNA was then fragmented, library-prepared and paired-end sequenced (NextSeq 500, Illumina Denmark ApS, Denmark) to a read length of 150 base pairs (bp). Raw sequence data are uploaded to the European Nucleotide Archive with accession no. PRJEB41153.

The obtained data were trimmed (quality threshold at 20 and minimum length of 50 bp) and mapped against reference sequences by the k-mer alignment (KMA, version 1.2.3t) method [30]. The reference database (genomic2_20191017) encompassed approximately 19 million sequences from bacteria, plasmids, archaea, viruses, fungi, protozoa and humans that were selected from NCBI GenBank assembly_summary.txt files where annotated tags were version_status = 'latest', genome_rep = 'Full', assembly_level = 'Complete genome' or 'Chromosome' and the human microbiome from 2014-07-07 was downloaded from https://ftp.ncbi.nlm.nih.gov/genomes/HUMAN_MICROBIOM/Bacteria. The pure cultures were processed in the same fashion from DNA extraction through to trimming. The paired-end reads were then assembled using Spades (version 3.7.0) [31]. Assembled contigs ≥1000 bp were added to the reference database.

## Relating target organism to number of sick people

The released amount at target organism in terms of CFU can be used to estimate the corresponding number of sick people (N). The aim is to calculate the number of sick people required in order to be able to detect the target organism with a specified probability. It can be done by either considering all samples that have been sequenced or only looking at the sensitivity in one sample where the target organism is at the highest peak. Below the total number

of reads in all samples is indicated with 'R' and the total number of summed target reads is indicated by 'r'.

Given the total amount of released target organism in terms of colony forming units (CFU) and the assumptions that a sick person delivers 200g faeces and the concentration of the pathogen is $10^6$ CFU/g, the "number of sick people" (N) contributing to the release is calculated in Eq 1.

$$N = \frac{target\ organism\ released[CFU]}{10^6 \left[\frac{CFU}{g}\right] \cdot 200[g]}$$

Eq 1

Given the total number of summed target reads (r) and the total number of summed reads in all samples (R) the probability that a sequenced read is from the target organism is calculated in the Eq 2 (P(A)).

$$P(A) = \frac{r}{R}$$

Eq 2

Given the assumption that there is a linear relation between the number of reads from the target organism in a sample and the number of sick people, the probability that a read in the sample is from the target organism released by one sick person (P(B)) can be estimated according to Eq 3.

$$P(B) = \frac{P(A)}{N}$$

Eq 3

In a surveillance situation, the aim of monitoring is to detect the presence of a hazard of interest in the sample, in this case DNA from an infectious pathogen. The likelihood that a sequenced read in a sample is from the target organism (P(C)) depends on how many sick persons (n) is releasing into the sample, and the likelihood that reads from these persons are from the target organism. Estimation of P(C) is done using the binomial formula (Eq 4) wherein (1-P(B)) is the likelihood that a read from one sick person is not the target read, and (1-P(B))$^n$ is the likelihood that a read is not the target read given n sick persons releasing into the sample, and 1 minus this likelihood is the likelihood that the read is from the target organism.

$$P(C) = 1 - (1 - P(B))^n$$

Eq 4

Given sequencing depth R, the likelihood that at least one read is from the target organism can be calculated using the binomial formula in Eq 5

$$P(D) = 1 - (1 - P(C))^R$$

Eq 5

By holding R fixed, the likelihood of detection at least one read from the target organism, was calculated for n sick people, ranging n from 1 to 10^2.5 (316) sick persons realising the target organism into the sewage system. The results were plotted (n sick people ~ probability of detection) and the number of sick people resulting into 95% likelihood for detection was approximated from the curves, each curve represent one or more metagenomic samples that have been mapped against a reference genome database which include the target organism of interest. In our setup using *S. hyicus* as target organism.

## Results

*S. hyicus* was detected on both release days and each subsample (A and B) as shown in S3 and S4 Tables in S1 File, including the sequencing depths for all samples. The average sequencing depths in terms of sample fragment counts are 27,298,791 and 23,831,412 for release day 1 and

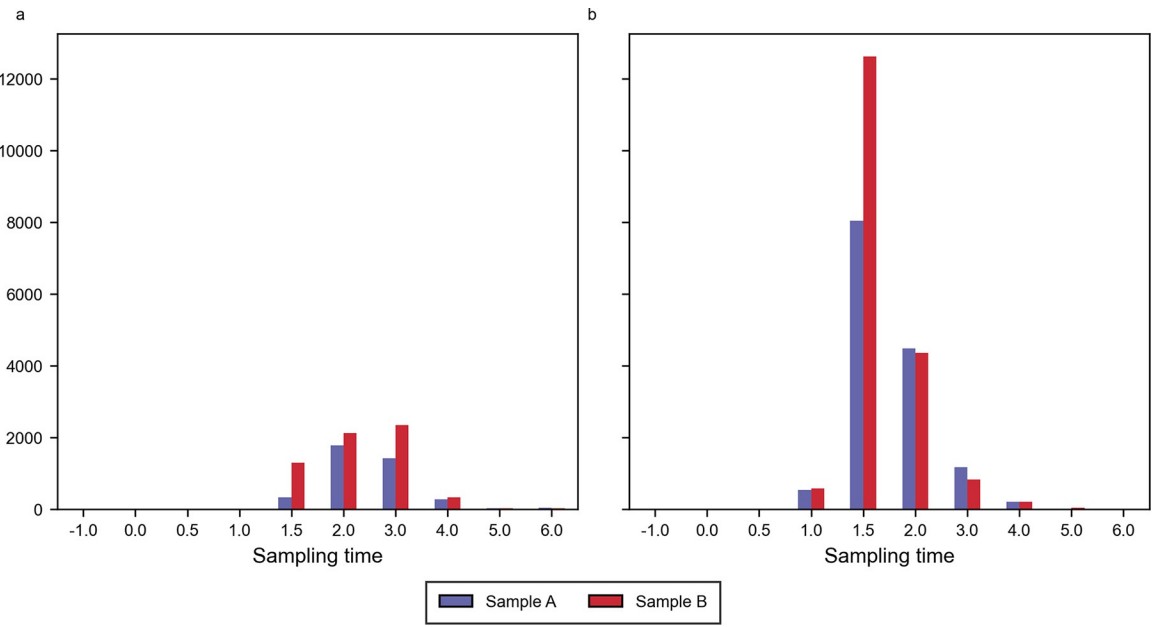

**Fig 1.** Staphylococcus hyicus fragment counts on release day 1 (a) and 2 (b).

2, respectively. The retrieved *S. hyicus* fragments were matched to the reference library with an average of 99.89 percent identity (range 98.56 to 100 percent) and a median p-value of $1\times10^{-26}$ (mean $1.99\times10^{-4}$; range $2.7\times10^{-2}$ to $1\times10^{-26}$). The 20 assembled reference contigs had a mean length of 123,972 bp (median 362,226 bp; range 1483 to 722,969 bp) and a sum of 2,479,435 bp. All raw mapping data is provided as supplementary materials.

On release day 1, the bacterial target passed the sampling point starting from 1.5 hours post release and peaked around 2-3h after release (Fig 1). On release day 2 a pronounced peak was observed 1.5h after release (Fig 1). The peak represented fragment counts 5–6 times higher compared to release day 1.

## Minimum number of sick people

Using Eqs 1–5 we are able to calculate the minimum number of sick people using summed fragment count data from the entire peak where *S. hyicus* are observed (Table 2). The probability distribution of the summed fragment counts for day 1 and 2 are shown in Fig 2.

**Table 2. Calculations of minimum number of sick people required to observe at least one from the waste water analysis.**

|  | Day 1 | | Day 2 | |
|---|---|---|---|---|
| **Subsample suffix** | **A** | **B** | **A** | **B** |
| Released amount (N) | 83,146 | | 95,525 | |
| Total fragments (R) | 134,564,123 | 173,057,637 | 178,577,988 | 173,512,546 |
| *S. hyicus* fragments (r) | 3,907 | 6,182 | 14,494 | 18,667 |
| Minimum amount required (n) | 64 | 41 | 20 | 16 |

Letters in parenthesis in the first column are: (N) Corresponding number of sick people as estimated from released amount of S. hyicus; (R) Summed number of all fragments in samples where S. hyicus was observed; (r) Summed number of S. hyicus fragments in same samples as those used to calculate (R); (n) minimum number of sick people that is required in order to observe at least one sick person with a probability of 95%.

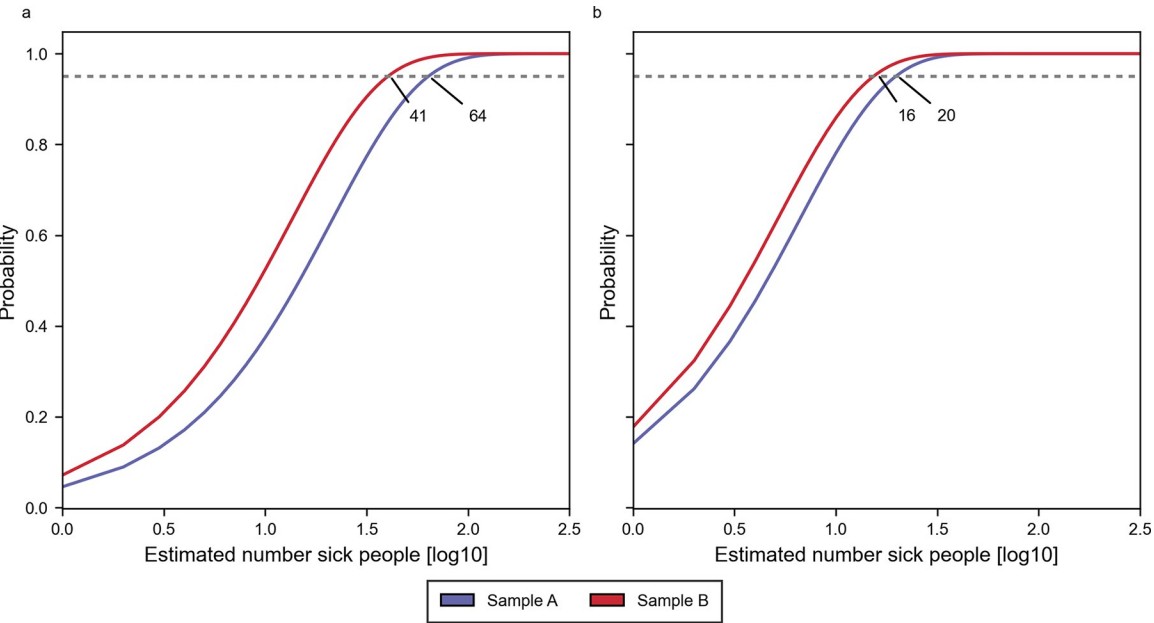

**Fig 2. Binomial distribution using fragment counts all Staphylococcus hyicus samples.** The figure show the plot of Eq 5 where fragment counts R and r (Eq 2) are summed from all samples where S. hyicus are identified. a and b are for release days 1 and 2, respectively. The horizontal dashed line represents a 95% probability threshold and at intersecting points are the minimum number of people required to observe a sick person i.e. 41, 64 on day 1 and 16, 20 on day 2 for subsamples A and B in blue and red color.

Using only fragment counts from a single sample where we observe the peak concentration of *S. hyicus* provide worse sensitivity for day 1, but for day 2 we find the sensitivity being equal for using either summed or max *S. hyicus* fragment counts (Fig 3, Table 3).

## Discussion

In this study we attempted to provide insight into the sensitivity of using metagenomics analyses of urban sewage for detection of people with diarrhea, using the release of known quantities of *S. hyicus* as a proxy. The experiment was replicated on two days a week apart and on both days, we could detect the target organism. We did observe an increased sensitivity on day 2 compared to day 1 both when looking at a single sample with the maximum number of observed *S. hyicus* fragments and when summing read counts from all samples where *S. hyicus* was observed. The increased average sensitivity on day 2 in terms on 'Minimum amount required (n)' is 3 and 4 based on data from Tables 2 and 3, respectively. We believe the sensitivity difference as a simple fact of an increased fraction of *S. hyicus* fragments on day 2 compared to day 1, where we see an r/R ratio i.e. P(A) being 3 and 7 based on data from Tables 2 and 3, respectively. If we look at P(B) values i.e. P(A) normalized by amount of released *S. hyicus* then we see an increased sensitivity on day 2 with a factor of 3 and 6, based on data in Tables 2 and 3, respectively. In conclusion the increased sensitivity of 3 on day 2 (All samples) can be explained by either P(A) or P(B) ratio's. The increased sensitivity of a factor of 4 on day 2 when only using data from a single peak with maximum amount of *S. hyicus* can roughly be explained by either P(A) of P(B) ratio's which can be calculated to a factor of 7 and 6, based on data in Table 3.

Previous studies using qPCR or cell cultures have estimated that sewage based surveillance for poliovirus, norovirus and SARS-CoV-2 could have a sensitivity of 1 infected person in 10,000 to 100,000 people [21, 32, 33]. From Tables 2 and 3 we find the 'Minimum amount

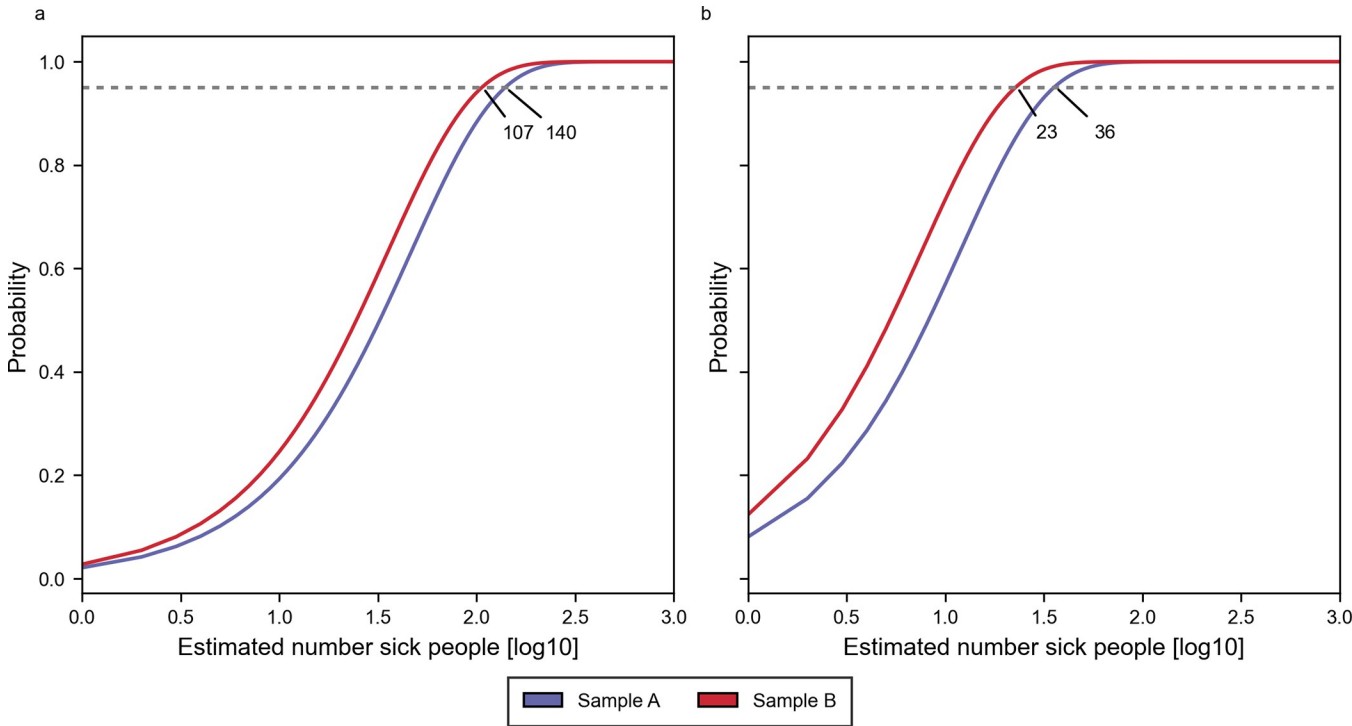

**Fig 3. Binomial distribution using fragment counts for highest Staphylococcus hyicus sample.** The figure show the plot of Eq 5 where fragment counts R and r (Eq 2) are from one samples where max S. hyicus was identified. a and b are for release days 1 and 2, respectively. The horizontal dashed line represents a 95% probability threshold and at intersecting points are the minimum number of people required to observe a sick person i.e. 107, 140 on day 1 and 23, 36 on day 2 for subsamples A and B in blue and red color.

required' being in the range from 16–140 i.e. number of sick people required in order to observe at least one with a probability of 95%. This translates into that we can identify 1 infected person among 36,000/140 = 257 to 36,000/16 = 2,250 people when normalized to the size of the WWTP serving approximately 36,000 people. Thus, considerably less than for virus using qPCR or cell culture. The sensitivity could naturally be increased if each sample is sequenced with more reads, but that would also come with additional costs. Future use of metagenomics for surveillance of pathogens would not only be for detection of novel outbreaks but also for measuring potential increased above an endemic background. This would require that the metagenomics approach is capable of detecting significant changes, thus putting further requirement on sensitivity. Caution should however, be taken when interpreting

**Table 3. Calculations of minimum number of sick people required to observe at least one from the waste water analysis, when the fragment counts is at peak.**

| Subsample suffix | Day 1 | | Day 2 | |
|---|---|---|---|---|
| | **A** | **B** | **A** | **B** |
| Released amount (N) | 83,146 | | 95,525 | |
| Total fragments (R) | 23,789,122 | 32,137,050 | 17,807,610 | 22,256,210 |
| *S. hyicus* fragments (r) | 1,784 | 2,344 | 8,052 | 12,625 |
| Minimum amount required (n) | 140 | 107 | 36 | 23 |

Letters in parenthesis in the first column are: (N) Corresponding number of sick people as estimated from released amount of S. hyicus; (R) Number of fragments in one sample where max S. hyicus fragments was at observed; (r) number of S. hyicus fragments in same sample as the one used to calculate (R); (n) minimum number of sick people that is required in order to observe at least one sick person with a probability of 95%.

our results since we used a very large release to estimate a theoretical sensitivity and additional studies should be conducted using release amounts closer to the theoretical sensitivity.

While use of metagenomics for continuous surveillance of sewage for pathogens clearly has great promises further studies are needed to establish per bacteria thresholds and sensitivities. This could include further release experiments with other bacterial species, as well as epidemiological studies. Most likely, such thresholds would have to be established individually for different sewer systems.

## Conclusions

We successfully demonstrated that we could detect a bacterial agent released into a sewage system using metagenomics analyses. Our current conservative estimate is that metagenomics analyses with 20 million reads would be able to detect a diarrheal pathogen if excreted from 41–140 and 16–36 sick people in the population of approximately 36,000 people at release days 1 and 2, respectively. This corresponds to a sensitivity of 1 in 257 to 2,250 people when normalizing with respect to population served by the WWTP.

## Supporting information

**S1 File. S1-S4 Tables.**
(DOCX)

**S2 File. S1-S9 Figs.**
(PDF)

## Acknowledgments

We thank our DTU staff Jacob Dyring Jensen, Natacha C. S. Veiergang, Claus Asperud Reesbøl, Fermina Petersen, Mildred A. Bautista and Danny Darby for their extraordinary technical assistance; the team at Usserød Renseanlæg for their welcome and enduring support; Kaare Mølbak from Statens Serum Institut for first suggesting the release of bacteria into a sewage system.

## Author Contributions

**Conceptualization:** Thomas N. Petersen, Frank M. Aarestrup.

**Formal analysis:** Simon Kohle, Thomas N. Petersen, Håkan Vigre, Markus Hans Kristofer Johansson.

**Funding acquisition:** Frank M. Aarestrup.

**Investigation:** Simon Kohle, Frank M. Aarestrup.

**Methodology:** Simon Kohle, Thomas N. Petersen, Markus Hans Kristofer Johansson.

**Project administration:** Thomas N. Petersen, Frank M. Aarestrup.

**Resources:** Frank M. Aarestrup.

**Supervision:** Thomas N. Petersen.

**Writing – original draft:** Simon Kohle.

**Writing – review & editing:** Thomas N. Petersen, Håkan Vigre, Markus Hans Kristofer Johansson, Frank M. Aarestrup.

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
