## [Decision Letter · Decision Letter 0]

24 Jan 2024

PONE-D-23-32545Metagenomic Analysis of Sewage for Surveillance of Bacterial Pathogens: A Release Experiment to Determine SensitivityPLOS ONE

Dear Dr. Aarestrup,

Thank you for submitting your manuscript to PLOS ONE. After careful consideration, we feel that it has merit but does not fully meet PLOS ONE’s publication criteria as it currently stands. Therefore, we invite you to submit a revised version of the manuscript that addresses the points raised during the review process.

We look forward to receiving your revised manuscript.

Kind regards,

Marwan Osman

Academic Editor

PLOS ONE

Journal Requirements:

3. For studies reporting research involving human participants, PLOS ONE requires authors to confirm that this specific study was reviewed and approved by an institutional review board (ethics committee) before the study began. Please provide the specific name of the ethics committee/IRB that approved your study, or explain why you did not seek approval in this case.

4. Please provide additional details regarding participant consent. In the ethics statement in the Methods and online submission information, please ensure that you have specified (1) whether consent was informed and (2) what type you obtained (for instance, written or verbal, and if verbal, how it was documented and witnessed). If your study included minors, state whether you obtained consent from parents or guardians. If the need for consent was waived by the ethics committee, please include this information.

6. Thank you for stating the following financial disclosure: "Main funder is the European Union"

7. Thank you for stating the following in the Acknowledgments Section of your manuscript: "This study was supported by the European Union’s Horizon 2020 research and innovation programme under grant agreement no. 874735 (VEO) and The Novo Nordisk Foundation (NNF16OC0021856: Global Surveillance of Antimicrobial Resistance).

Please remove any funding-related text from the manuscript and let us know how you would like to update your Funding Statement. Currently, your Funding Statement reads as follows: "Main funder is the European Union"

9. We note that Figure S5 in your submission contain [map/satellite] images which may be copyrighted. All PLOS content is published under the Creative Commons Attribution License (CC BY 4.0), which means that the manuscript, images, and Supporting Information files will be freely available online, and any third party is permitted to access, download, copy, distribute, and use these materials in any way, even commercially, with proper attribution. For these reasons, we cannot publish previously copyrighted maps or satellite images created using proprietary data, such as Google software (Google Maps, Street View, and Earth). For more information, see our copyright guidelines: http://journals.plos.org/plosone/s/licenses-and-copyright.

a. You may seek permission from the original copyright holder of Figure S5 to publish the content specifically under the CC BY 4.0 license.  

Reviewers' comments:

Reviewer's Responses to Questions

**Comments to the Author**

1. Is the manuscript technically sound, and do the data support the conclusions?

Reviewer #1: Yes

Reviewer #2: Yes

2. Has the statistical analysis been performed appropriately and rigorously? 

Reviewer #1: Yes

Reviewer #2: Yes

3. Have the authors made all data underlying the findings in their manuscript fully available?

Reviewer #1: Yes

Reviewer #2: Yes

4. Is the manuscript presented in an intelligible fashion and written in standard English?

Reviewer #1: Yes

Reviewer #2: Yes

5. Review Comments to the Author

Reviewer #1: The manuscript aims to investigate the sensitivity of metagenomic sequencing for the detection of specific sewage bacteria in the context of wastewater-based epidemiology. This was conducted in situ by spiking known quantities of extraneous bacteria (Staphyloccus hyicus) into a sewer system before metagenomic sequencing of samples from the associated wastewater treatment plant. The authors highlight the potential for wastewater metagenomics to estimate the number of sick individuals with an increased sensitivity over previous non-metagenomic methods.

Overall, I found the manuscript to be well written, interesting, and novel as relatively few published studies have performed spiking/spot-release experiments in the context of wastewater-based epidemiology. However, I do believe clarification of some points, potential further analyses and additional discussion would enhance the manuscript.

I have following specific comments.

1. Line 137-140: Since these initial values are pivotal to the overall experiment and calculations of sensitivity, please provide more detailed justification on why a 200 g feces assumption was made (supporting reference?) and the same for CFU per gram - is based on observed amounts from Campylobacter spp. and Salmonella ssp infection stool samples? If so, please provide supporting reference

2. Line 220-222: The metagenomic sequencing depth is not clearly reported but can be partially inferred through the reported total fragments in Tables 1 and 2 (Line 299-311) or from the discussion (Line 328). This is of crucial importance and relevance to the manuscript analyses since sensitivity to detect specific targets in metagenomes is directly tied to metagenomic sequencing depth and sample complexity (https://doi.org/10.1186/s40793-019-0347-1). Please report the sequencing depth achieved per sample and discuss the impact of sequencing depth as a limitation (i.e. deeper sequencing would likely be required due to reduced/scattered real world release as compared to the large experimental spot-release)

3. Line 294-317: This section is very interesting and highlights potential sensitivity differences between results based on multiple samples and a single peak sample (as seen in other studies), but this is not currently mentioned in the discussion. Was there any factor which could explain the reduced sensitivity in Day 1-peak conc (e.g. weather, flow, taxonomic composition apart from S. hyicus). It would also be interesting to compare the sensitivities for all single samples and not just the sample with peak S. hyicus concentration

4. Line 320-343: The discussion section is very limited in scope and does not address several presented results which warrant discussion such as the similarities and differences between sensitivity of compound sample results (Table 1) and peak conc. (Table 2) – as above.

5. Line 329: The calculation of the value “1 in 260 to 2,250 people” is not clearly described. It can be inferred that it originates from from Tables 1 and 2 using Minimum amount required (n) and normalised by size of the WWTP through the conclusion only (Line 349-351). Please clarify at first mention.

Minor comments/typographical issues:

Line 32: Shotgun is more commonly accepted over “shot-gun” – please update

Line 139: Please update citation format to be consistent with numbering used throughout

Line 226: Please reference where the sequences were downloaded from to generate the reference database “genomic2_20191017”

Line 230: Please provides reference for Spades as done for KMA (Line 225-226)

Line 336: Typo “Cation” should be Caution

Reviewer #2: The manuscript, entitled "Metagenomic Analysis of Sewage for Surveillance of Bacterial Pathogens: A Release Experiment to Determine Sensitivity", provided new information about the implication of metagenomics for detecting pathogens from sewage. The manuscript is well-written and organized.

The results can be applied to monitoring bacterial pathogens.

For the methods, could you please explain the reason for using only the CFU of 1.66 and 1.91 but not different lower concentrations?

The second question is about the real sample from the community. The experiment revealed the sensitivity of the method for tracing the pathogens. However, is the method good for the real situation when the community has the patients caused by other pathogens? It would be more effective if the authors applied this method to a specific pathogen.

Thank you.

6. PLOS authors have the option to publish the peer review history of their article (what does this mean?). If published, this will include your full peer review and any attached files.

Reviewer #1: No

Reviewer #2: No

---

## [Author Response · Author response to Decision Letter 0]

12 Feb 2024

PONE-D-23-32545

Metagenomic Analysis of Sewage for Surveillance of Bacterial Pathogens: A Release Experiment to Determine Sensitivity

PLOS ONE

Comments from the editor:

6. Thank you for stating the following financial disclosure: "Main funder is the European Union"

7. Thank you for stating the following in the Acknowledgments Section of your manuscript: "This study was supported by the European Union’s Horizon 2020 research and innovation programme under grant agreement no. 874735 (VEO) and The Novo Nordisk Foundation (NNF16OC0021856: Global Surveillance of Antimicrobial Resistance).

Please remove any funding-related text from the manuscript and let us know how you would like to update your Funding Statement. Currently, your Funding Statement reads as follows: "Main funder is the European Union"

Has been included in the cover letter

Has been included

9. We note that Figure S5 in your submission contain [map/satellite] images which may be copyrighted. All PLOS content is published under the Creative Commons Attribution License (CC BY 4.0), which means that the manuscript, images, and Supporting Information files will be freely available online, and any third party is permitted to access, download, copy, distribute, and use these materials in any way, even commercially, with proper attribution. For these reasons, we cannot publish previously copyrighted maps or satellite images created using proprietary data, such as Google software (Google Maps, Street View, and Earth). For more information, see our copyright guidelines: http://journals.plos.org/plosone/s/licenses-and-copyright.

I have deleted the figure and changed the numbering accordingly

Reviewer comments 

Reviewers' comments:

Reviewer's Responses to Questions

Comments to the Author

1. Is the manuscript technically sound, and do the data support the conclusions?

Reviewer #1: Yes

Reviewer #2: Yes

2. Has the statistical analysis been performed appropriately and rigorously? 

Reviewer #1: Yes

Reviewer #2: Yes

3. Have the authors made all data underlying the findings in their manuscript fully available?

Reviewer #1: Yes

Reviewer #2: Yes

4. Is the manuscript presented in an intelligible fashion and written in standard English?

Reviewer #1: Yes

Reviewer #2: Yes

5. Review Comments to the Author

Reviewer #1: The manuscript aims to investigate the sensitivity of metagenomic sequencing for the detection of specific sewage bacteria in the context of wastewater-based epidemiology. This was conducted in situ by spiking known quantities of extraneous bacteria (Staphyloccus hyicus) into a sewer system before metagenomic sequencing of samples from the associated wastewater treatment plant. The authors highlight the potential for wastewater metagenomics to estimate the number of sick individuals with an increased sensitivity over previous non-metagenomic methods.

Overall, I found the manuscript to be well written, interesting, and novel as relatively few published studies have performed spiking/spot-release experiments in the context of wastewater-based epidemiology. However, I do believe clarification of some points, potential further analyses and additional discussion would enhance the manuscript.

I have following specific comments.

1. Line 137-140: Since these initial values are pivotal to the overall experiment and calculations of sensitivity, please provide more detailed justification on why a 200 g feces assumption was made (supporting reference?) and the same for CFU per gram - is based on observed amounts from Campylobacter spp. and Salmonella ssp infection stool samples? If so, please provide supporting reference.

THERE IS AS FAR AS WE CAN TELL A SURPRISINGLY LIMITED NUMBER OF STUDIES THAT HAVE QUANTIFIED AMOUNT OF FECES EXCRETED BY HEALTH AND DIARRHEAL PEOPLE AND ALSO VERY LIMITED STUDIES QUANTIFYING THE AMOUNT OF PATHOGENS. WE HAVE INCLUDED SOME REFERENCES, BUT THIS CERTAINLY WARRENTS FURTHER STUDIES.

2. Line 220-222: The metagenomic sequencing depth is not clearly reported but can be partially inferred through the reported total fragments in Tables 1 and 2 (Line 299-311) or from the discussion (Line 328). This is of crucial importance and relevance to the manuscript analyses since sensitivity to detect specific targets in metagenomes is directly tied to metagenomic sequencing depth and sample complexity (https://doi.org/10.1186/s40793-019-0347-1). Please report the sequencing depth achieved per sample and discuss the impact of sequencing depth as a limitation (i.e. deeper sequencing would likely be required due to reduced/scattered real world release as compared to the large experimental spot-release)

Added average sequencing depth in results section and the individual sample depths are in S3 and S4.

3. Line 294-317: This section is very interesting and highlights potential sensitivity differences between results based on multiple samples and a single peak sample (as seen in other studies), but this is not currently mentioned in the discussion. Was there any factor which could explain the reduced sensitivity in Day 1-peak conc (e.g. weather, flow, taxonomic composition apart from S. hyicus). It would also be interesting to compare the sensitivities for all single samples and not just the sample with peak S. hyicus concentration

We have looked at the data trying to explain the increased sensitivity on day 2 which was 3 and 4 times higher, using either all fragment count data (table 2) or only fragment counts from the single sample with maximum S. hyicus (table 3). It seems that the observed abundance of S. hyicus in the samples is a major factor i.e. P(A). We did not do it for all single samples except for the one with maximum amount of observed S. hyicus. The explanation is elaborated in Discussion section.

4. Line 320-343: The discussion section is very limited in scope and does not address several presented results which warrant discussion such as the similarities and differences between sensitivity of compound sample results (Table 1) and peak conc. (Table 2) – as above.

Much alike the above explanation we also tried to explain the sensitivity difference in terms of P(B) which also include the amount of released S. hyicus in terms of the approximated ‘number of sick people N’. Using P(B) was slightly better at explaining the sensitivity differences compared to the P(A) factor. This is also elaborated in Discussion section.

5. Line 329: The calculation of the value “1 in 260 to 2,250 people” is not clearly described. It can be inferred that it originates from from Tables 1 and 2 using Minimum amount required (n) and normalised by size of the WWTP through the conclusion only (Line 349-351). Please clarify at first mention.

This has been clarified in the discussion where we show the formula used in the calculations.

Minor comments/typographical issues:

Line 32: Shotgun is more commonly accepted over “shot-gun” – please update

This is now corrected

Line 139: Please update citation format to be consistent with numbering used throughout

We have now included the reference

Line 226: Please reference where the sequences were downloaded from to generate the reference database “genomic2_20191017”

This is now explained in the methods section under ‘DNA-sequencing and metagenomic analyses’

Line 230: Please provides reference for Spades as done for KMA (Line 225-226)

For Frank: added full reference in the end of manuscript and just wrote (Nurk et al. 2017) in text but that needs to be formatted correctly

Line 336: Typo “Cation” should be Caution

Now corrected

Reviewer #2: The manuscript, entitled "Metagenomic Analysis of Sewage for Surveillance of Bacterial Pathogens: A Release Experiment to Determine Sensitivity", provided new information about the implication of metagenomics for detecting pathogens from sewage. The manuscript is well-written and organized.

THANK YOU

The results can be applied to monitoring bacterial pathogens.

THANK YOU

For the methods, could you please explain the reason for using only the CFU of 1.66 and 1.91 but not different lower concentrations?

IT WAS PURELY A QUESTION OF RESSOURCES. WE AGREE THAT ADDITIONAL EXPERIMENTS SHOULD BE DONE, BUT THAT WILL NOT CHANGE THE MAIN POINT, NAMELY THAT THE SENSITIVITY CAN BE LOW.

The second question is about the real sample from the community. The experiment revealed the sensitivity of the method for tracing the pathogens. However, is the method good for the real situation when the community has the patients caused by other pathogens? It would be more effective if the authors applied this method to a specific pathogen.

AGREED. HOWEVER, RELEASING FORT EXAMPLE A PATHOGENIC SALMONELLA BACTERIUM WOULD COME WITH BOTH ETHICAL AND TECHNICAL CONCERNS. RELEASING A KNOWN PATHOGEN COULD BE CONSIDERED A HUMAN HEALTH RISK AND FURTHERMORE, ANY SIGNALS FROM THE RELEASED ORGANISM COULD BE MIXED WITH NATURALLY OCCURING INFECTIONS.

IN THIS CASE WE DECIDED ON A PROVEN NON-PATHOGENIC ANIMAL ASSOCIATED BACTERIAL SPECIES TO LIMIT HUMAN HEALTH RISKS AND ALSO A BACTERIAL THAT HAVE NOT BEEN DETECTED IN ANY SEWAGE WE HAVE PREVIOUSLY ANALYSED.

Thank you.

---

## [Decision Letter · Decision Letter 1]

5 Mar 2024

Metagenomic Analysis of Sewage for Surveillance of Bacterial Pathogens: A Release Experiment to Determine Sensitivity

PONE-D-23-32545R1

Dear Dr. Aarestrup,

We’re pleased to inform you that your manuscript has been judged scientifically suitable for publication and will be formally accepted for publication once it meets all outstanding technical requirements.

Kind regards,

Marwan Osman

Academic Editor

PLOS ONE

Reviewers' comments:

Reviewer's Responses to Questions

**Comments to the Author**

1. If the authors have adequately addressed your comments raised in a previous round of review and you feel that this manuscript is now acceptable for publication, you may indicate that here to bypass the “Comments to the Author” section, enter your conflict of interest statement in the “Confidential to Editor” section, and submit your "Accept" recommendation.

Reviewer #1: All comments have been addressed

2. Is the manuscript technically sound, and do the data support the conclusions?

Reviewer #1: (No Response)

3. Has the statistical analysis been performed appropriately and rigorously? 

Reviewer #1: (No Response)

4. Have the authors made all data underlying the findings in their manuscript fully available?

Reviewer #1: (No Response)

5. Is the manuscript presented in an intelligible fashion and written in standard English?

Reviewer #1: (No Response)

6. Review Comments to the Author

Reviewer #1: (No Response)

7. PLOS authors have the option to publish the peer review history of their article (what does this mean?). If published, this will include your full peer review and any attached files.

Reviewer #1: **Yes: **Kevin K Chau

---

## [Editor Report · Acceptance letter]

4 May 2024

PONE-D-23-32545R1 

PLOS ONE

Dear Dr. Aarestrup, 

I'm pleased to inform you that your manuscript has been deemed suitable for publication in PLOS ONE. Congratulations! Your manuscript is now being handed over to our production team.

Kind regards, 

on behalf of

Dr. Marwan Osman 

Academic Editor

PLOS ONE